# Highly Porous Hydroxyapatite/Graphene Oxide/Chitosan Beads as an Efficient Adsorbent for Dyes and Heavy Metal Ions Removal

**DOI:** 10.3390/molecules26206127

**Published:** 2021-10-11

**Authors:** Nguyen Van Hoa, Nguyen Cong Minh, Hoang Ngoc Cuong, Pham Anh Dat, Pham Viet Nam, Pham Hau Thanh Viet, Pham Thi Dan Phuong, Trang Si Trung

**Affiliations:** 1Faculty of Food Technology, Nha Trang University, Nha Trang 650000, Vietnam; datpa@ntu.edu.vn (P.A.D.); danphuong@ntu.edu.vn (P.T.D.P.); trungts@ntu.edu.vn (T.S.T.); 2Institute for Biotechnology and Environment, Nha Trang University, Nha Trang 650000, Vietnam; minhnc@ntu.edu.vn; 3Faculty of Biotechnology, Binh Duong University, Thu Dau Mot 55000, Vietnam; hncuong@bdu.edu.vn; 4Faculty of Fishery, Ho Chi Minh City University of Food Industry, Ho Chi Minh City 70000, Vietnam; nampv.fisheries@gmail.com; 5Faculty of Chemistry, Da Lat University, Dalat 670000, Vietnam; vietpht@dlu.edu.vn

**Keywords:** hydroxyapatite, graphene oxide, chitosan, beads, absorbent, wastewater treatment

## Abstract

Dye and heavy metal contaminants are mainly aquatic pollutants. Although many materials and methods have been developed to remove these pollutants from water, effective and cheap materials and methods are still challenging. In this study, highly porous hydroxyapatite/graphene oxide/chitosan beads (HGC) were prepared by a facile one-step method and investigated as efficient adsorbents. The prepared beads showed a high porosity and low bulk density. SEM images indicated that the hydroxyapatite (HA) nanoparticles and graphene oxide (GO) nanosheets were well dispersed on the CTS matrix. FT-IR spectra confirmed good incorporation of the three components. The adsorption behavior of the obtained beads to methylene blue (MB) and copper ions was investigated, including the effect of the contact time, pH medium, dye/metal ion initial concentration, and recycle ability. The HGC beads showed rapid adsorption, high capacity, and easy separation and reused due to the porous characteristics of GO sheets and HA nanoparticles as well as the rich negative charges of the chitosan (CTS) matrix. The maximum sorption capacities of the HGC beads were 99.00 and 256.41 mg g^−1^ for MB and copper ions removal, respectively.

## 1. Introduction

Water pollution is a global environmental problem that affects human beings and aquatic ecology [1,2]. The contamination of pure water resources, including rivers, lakes, and oceans, is mainly due to human activities. Among them, dyes and heavy metals are two of the most severe aquatic pollutants [1]. These pollutants usually come from effluents of various industries such as textile dyeing, printing, cosmetics, paints, leather products, mining, chemical manufacturing, electroplating, battery manufacturing, and metal processing [3,4,5]. For example, methylene blue (MB) is a popular thiazine dye used in silk, cotton, and wood dyeing processing. Its release into the environment is hazardous for all living beings when in direct contact with it. For humans and animals, MB may cause eye burns, inhalation, irritation to the gastrointestinal tract, nausea, vomiting, mental confusion, and diarrhea [6,7]. In addition to organic dyes, heavy metal ions from effluents also cause serious environmental problems for plants and animals. The human body may get chronic diseases and even cause organ failure by taking a certain amount of heavy metal ions [8,9]. Among them, copper (Cu) is a common metal contaminant in food such as shellfish, liver, mushrooms, nuts, and chocolate, which may cause serious health issues, such as acute gastroenteritis, jaundice, memory deficit, and liver toxicity [10,11]. Therefore, it is crucial to remove heavy metal ions and dyes from water.

There are various techniques available for the removal of heavy metals and dyes from wastewater, such as membrane filtration [8,12], chemical precipitation [13], ion exchange [14], electrolytic techniques [15], biological methods [16], chemical oxidation or reduction process [17], and adsorption [4,5,6,7,9,10,11]. The adsorption method has been exhibited as a facile, efficient, low-cost, and environment-friendly method for wastewater treatment. Tremendous efforts have been paid to design and fabricate various adsorbents for dye and heavy metals in practice. However, the development of excellent adsorption materials with cost-effective and eco-friendly properties is required.

Chitosan (CTS), a deacetylated product of chitin, was mainly commercially produced from custance shells. It has valuable properties, such as biocompatibility, biodegradability, and antimicrobial activity [18]. CTS has been reported as an efficient adsorbent for heavy metal and dye removal from wastewater due to amino (−NH_2_) and hydroxyl (–OH) groups in its chemical structure that serve as coordination sites [18,19]. Various CTS-based composites have also been developed to remove pollutants from polluted water [3,4,9,11,18,19,20,21]. However, the CTS chains tend to form dense structures, resulting in a reduction in their surface area and adsorption efficiency. Therefore, it needs to combine with the high surface area materials.

Hydroxyapatite (HA, Ca_5_(PO_4_)_3_(OH)) is a non-toxic inorganic compound which naturally occurs in human and animal bone. Moreover, the Ca (II) ions in the HA structure can be exchanged easily with similar metallic cations, such as Pb(II), Cd(II), Cu(II), Zn(II), and Fe(III) [22,23,24]. Therefore, HA-based materials have been used as an efficient adsorbent for metal ion removal. Besides, the nano-HA powder was indicated to interact with chitosan structures effectively, forming nanocomposites with enhanced properties for various biomedical and environmental applications [20,21].

This study prepared a series of highly porous adsorbents by combining high surface area GO nanosheets with two biomaterials, HA and CTS, at room temperature. The HA and CTS were recovered from fishbone and shrimp shells, respectively. The prepared beads were characterized for morphology and physic-chemical structure by SEM, FT-IR, and BET measurements. As adsorbents, the products were used to remove methylene blue and Cu(II) ions from aqueous solutions under varying adsorption time, pH medium, and initial pollutant concentration. In addition, various isothermal models and the adsorption kinetics were investigated. This report aimed to develop high capacity, eco-friendly, and low-cost adsorbents to remove methylene blue and copper ions from aqueous solutions.

## 2. Results and Discussion

### 2.1. Characterization of Prepared Beads

Figure 1 shows SEM images at the surface and cross-section morphology of HGC beads with various GO loadings at different magnifications. The HGC-0 sample without GO fraction presents a compact structure comprising small hydroxyapatite particles distributed in the chitosan matrix (Figure 1a,b). Even the HA particles are in a nano-size with a porous structure (Appendix A). However, after loading with a specific GO fraction, all HGC beads showed a high porosity (Figure 1c–h). At low magnifications, the surface of the beads was relatively smooth, but the cross-section with micropores can be observed (Figure 1c,e,g). At high magnification, the samples displayed a well-distribution of HA particles and GO sheets in the chitosan matrix (Figure 1d,f,h). Besides, the tiny pores could be seen in the HGC-2 and HGC-3 samples (Figure 1f,h). These porous structures enhance pollutant ion and molecule access and trapping into the beads and provide abundant active sites for the adsorption of pollutants.

Furthermore, the porous characteristics of the composite beads were measured by N_2_ adsorption/desorption isotherms and presented in Table 1. The specific BET specific surface area of the beads increased as an increase of the GO loadings. This behavior can be attributed to the distribution of HA particles and CTS chains on the surface of GO sheets, which have multiple gaps among the layers. The more GO was added, the higher the surface area of the beads was observed. The highest area value of 29.74 m^2^ g^−1^ is obtained when the GO loading was 0.03 g. Besides, the pore sizes of prepared beads are about 10–16 nm, indicating the mesoporous structure in nature. This porous structure provides a large contact area between the absorbents and pollutants, enhancing absorption efficiency.

Figure 2 shows the XRD patterns of bare GO, bare HA, pure CTS, and various composite beads. The GO and CTS spectra have dominant peaks at around 12° and 20°, respectively; the HA spectrum has the same characteristic peaks of the reference pattern of HA No. JCPDS-09-0432 (Appendix A). All composite beads have the prominent peaks of HA and CTS fractions, suggesting that the crystal structure of HA did not change after combining with CTS and GO. Besides, none of the GO peaks was observed, which may be due to its small loadings and well-distribution in the composite structure.

The chemical structure of the samples was determined by FT-IR spectroscopy. Figure 3 presents the FT-IR spectra of the obtained composite beads. All spectra display usual characteristic stretching vibration bands for pure CTS (Appendix A), and pure HA (Appendix A). Practically, the broad absorption bands were centered at 3250 cm^−1^ (the stretching vibrations of –OH and N–H groups). The intensities were changed due to the water absorption on the surface of beads and the hydrogen bonds occurring in a crosslinked polymer network [25]. Other absorption bands were observed at 2913 cm^−1^ (symmetrical stretching vibrations of C–H groups), 1404 cm^−1^ (asymmetric stretching vibrations of COO^−^ groups), 1015 cm^−1^ (C–H and P–O groups), 560 cm^−1^ (P–O groups) [26]. The band was observed at 1549 cm^−1^, which was attributed to primary and secondary amide groups in the chitosan matrix [27]. Due to a small fraction of GO and the overlap of absorption bands, the characteristic bands of bare GO (Appendix A) don’t significantly affect the spectra of the composite beads. Overall, the beads’ spectra supported that chitosan chains were successfully combined with GO sheets and HA particles.

### 2.2. Effect of Contact Time

Figure 4 presents the effect of contact time on the adsorption capacities for the adsorption of MB (Figure 4a) and Cu (II) ions (Figure 4b) on various HGC bead structures. In both cases, the adsorption rates rapidly increased as time progressed due to the rapid transport of pollutants from the solution to the abundance active sites in the high porous beads. Afterward, the adsorption rates decreased and did not change significantly with the prolonging time, suggesting the saturation of the adsorption active sites in the bead structures. Therefore, it could be assumed that the adsorption equilibriums have occurred at 120 min for MB and 480 min for Cu (II) ions adsorption. The highest MB and Cu (II) adsorption capacities of the HGC-2 bead were 96.9 mg of MB for 120 min and 226 mg of Cu(II) ions per gram of dried beads for 480 min, respectively.

On the other hand, it showed that the adsorption capacity was significantly affected by the bead structures. The higher porosity of the beads was obtained, the higher the adsorption capacity was achieved. The HGC-2 exhibited the highest capacity among the tested samples, which could be attributed to the highest BET surface area and the even distribution of HA particles and GO sheets in the CTS matrix. Besides, the HGC-2 sample displayed many tiny pores in the composite structure, increasing the available active sites for adsorption and fast diffusion and transports of pollutants into the bead network.

### 2.3. pH Effect

The HGC beads consist of three components (GO, CTS, HA), which contributed to the MB and Cu (II) adsorption due to their functional groups (–NH_2_, –COOH, –OH) and their high porous surface. For example, the initial pH of the solution highly affects the adsorption process of chitosan due to the lone pair of electrons on the amine and carboxylic groups in its structure [28]. The amine groups are protonated partially or entirely at pH ≤ 6.5 (pKa of NH_2_ groups), forming –NH_3_^+^ cations, whereas the hydroxyl groups’ protonation occurs at pH ≤ 4.7 (pKa of –COOH groups). The CTS cationic form with positive charges easily attracts anionic compounds [29,30]. However, at a high pH medium, these amine groups begin to lose their charges due to an increase in OH^−^ groups, resulting in more negative on the surface of chitosan-based materials. These processes can be displayed in detail as follows [31]:

CTS-NH_2_ + H_3_O^+^ (acidic solution) ⇆ CTS-NH_3_^+^ (cationic form) + H_2_O

CTS-NH_2_ + H_2_O + OH^−^ (basic solution) ⇆ CTS-NH^−^ (anionic form) + 2H_2_O

Figure 5 presents the effect of pH medium on MB and Cu (II) adsorption in the range of 4.0–10.0 and 4.0–6.0, respectively. Generally, the adsorption capacity of the adsorbent increased with an increase in the initial pH values. Figure 5a shows an increase in the adsorption capacity of HGC beads on MB as a function of pH values. It may be explained that the surface charge of HGC bead changed from positive to negative with the increase of pH from about 4.0 to 10.0. Thus, the HGC anionic groups can interact with MB cationic groups, resulting in a significant increase in adsorption capacity. The maximum adsorption capacity is observed at about pH 8 and nearly unchanged at pH 10. For the adsorption of Cu(II) ions, the initial pH values were not investigated at higher than 6.0 since Cu(II) ions can be formed Cu(OH) or Cu(OH)_2_ at these pH values [29,30]. Figure 5b showed that the adsorption capacity of HGC beads toward Cu(II) ions slightly increased with an increase in pH values. At pH 4, a low adsorption capacity was observed due to the deprotonation of –NH_2_ groups in chitosan chains and the intense competition of H_3_O^+^ ions with Cu(II) ions for adsorption. Besides, electrostatic repulsion between Cu(II) ions and the –NH_3_^+^ groups may inhibit the Cu(II) ion adsorption [31]. At a high pH medium, the more negative of the composite surface enhanced the adsorption of cationic metal ions onto the negative-charge sites. However, the precipitation of copper hydroxides could partially occur at pH 6. Overall, the maximum adsorption capacities of the HGC-2 composite bead were 99.00 for MB adsorption and 256.41 mg g^−1^ for Cu (II) ions adsorption at the initial pH values of 8.0 and 5.0, respectively. These pH values were applied for further adsorption investigations.

### 2.4. Sorption Isotherms

Two sorption isotherms, including Langmuir and Freundlich isotherms, were used to investigate the equilibrium characteristics for MB and Cu(II) ion adsorption on the HGC-2 beads. The Langmuir model assumes a homogeneous surface with individual chemical adsorbents for physical adsorption (monolayer) within a low concentration range. Maximum adsorption was considered at the saturated monolayer of adsorbates formed on the surface of the adsorbent. The Freundlich isotherm is an empirical model for the adsorption of a single pollutant system at the high and middle concentrations on uneven surfaces of adsorbents. This model is not applicable for a low concentration of pollutants due to the requirements of Henry’s law [19]. The above two isotherm models are considered for practical applications by comparing the standard deviation between the experimental and modeled data. The adsorption capacities of HGC-2 were evaluated at different initial concentrations of MB (5–65 mg L^−1^) and Cu^2+^ (100–1000 mg L^−1^). Figure 6 presents the fitting Langmuir and Freundlich isotherms for MB (Figure 6a) and Cu(II) ions (Figure 6b) adsorption on the HGC-2 sample. In both cases, the higher the initial pollutant concentrations were tested, the higher adsorption capacities were achieved. Table 2 shows the isotherm parameters. The maximum adsorption capacities of MB and Cu(II) ions, obtained by Langmuir fitting, are 99.00 mg g^−1^ and 256.41 mg g^−1^, respectively. Based on the two models’ linear correlation coefficients (R^2^) and chi-square (χ^2^) values, the Langmuir model exhibited a better fit for MB and Cu(II) adsorption on the HGC-2 sample than the Freundlich model, indicating that monolayer adsorption can better describe the adsorption process.

### 2.5. Adsorption Kinetics

The pseudo-first-order and pseudo-second-order models are used to evaluate the adsorption kinetics. Figure 7 presents the linear plots of ln(q_e_–q_t_) vs. t (pseudo-first-order kinetic model) and t/q_t_ vs. t (pseudo-second-order kinetic model) for adsorption of MB and Cu(II) ions on HGC-2 beads. Table 3 shows the kinetic adsorption parameters. The pseudo-second-order kinetic model is better fitted to describe the interaction of the beads to both MB and Cu(II) ions, as demonstrated by the smaller SD values. Therefore, it may be assumed that the chemical reactions between dye or metal ions and the HGC beads are the primary interaction controlling the adsorption rate and process [26,31,32]. Furthermore, it reported that metal ions binding to active sites in the chitosan-based adsorbents formed both monolayer and multilayer, resulting in the rate-limiting adsorption process [26,32].

### 2.6. Cyclic Test

As the microbead form, the adsorbent can be easily separated and reused. Figure 8 shows the adsorption recycles of the HGC-2 beads for MB (Figure 8a) and Cu(II) ions (Figure 8b). For five cycles, nearly 100% adsorption capacities could be obtained for both MB and Cu(II) ions. The slight loss of capacities may be due to the remaining MB and Cu^2+^ ions in the beads. This phenomenon was also observed in previous works [33].

## 3. Materials and Methods

### 3.1. Materials

Graphene oxide (GO) was fabricated from graphite (99.995%, Alfa Aesar, Ward Hill, MA, USA) using a modified Hummers method [34]. Hydroxyapatite (HA, an average size of ca. 50 nm) and chitosan (CTS, M_w_ of 150 kDa, deacetylated degree of about 93%) were prepared from catfish bone and shrimp shells, respectively, in the laboratory using the optimum conditions of previous reports [35,36]. Sodium tripolyphosphate (TPP, >99%, Aldrich, St. Louis, MO, USA), CH_3_COOH (99.6%, Merck, Darmstadt, Germany), and other chemicals were of reagent grade and used as received.

### 3.2. Preparation of the HGC Beads

In a synthesis of the beads, chitosan (0.5 g) and HA (0.5 g) was dispersed in 50 mL acetic acid solution (1.0 vol.%) under sonication for 30 min. Then, GO (0.02 g) was added to the above suspension and sonicated for another 30 minutes. Next, the resulting solution was dropped into a mixture of 5.0% TPP solution and 4% NaOH solution at a 17: 3 (*v*/*v*) ratio under slowly stirring to give microparticles. Then, the particles were washed DI water and exchanged with absolute ethanol at 25 °C. Finally, the aerogel beads were obtained for characterization by freeze-drying at −80 °C. The received product was denoted as HGC-2. For the comparison, the samples were prepared using the same amount of HA (0.5 g) and CTS (0.5 g) with 0.01 and 0.03 g of GO, labeled as HGC-1, and HGC-3, respectively. Besides, the beads were prepared using 0.5 g of HA and 0.5 g of CTS without GO, named HGC-0.

### 3.3. Characterization

The chemical structure, crystallinity, and morphology of the samples were characterized by FTIR (Nicolet iS10, Thermo Scientific) within a range of 500–4000 cm^−1^ at a resolution of 16 cm^−1^ within 32 scans, XRD (PANalytical, X’Pert-PRO MPD, Almelo, The Netherlands) using Cu Kα radiation, SEM (Hitachi, S-4800, Tokyo, Japan). The surface area of samples was determined using the BET method at 77 K of liquid nitrogen. The sorbent was degassed at 473.15 K in a vacuum before measuring a Quantachrome Autosorb-1 Instrument (Boynton Beach, FL, USA). The samples’ specific surface area from the nitrogen adsorption isotherms was calculated using the Brunauer–Emmett–Teller (BET) equation.

### 3.4. Adsorption Test

The stock solutions of 100 mg L^−1^ MB or 2000 mg L^−1^ Cu (II) ions were prepared by dissolving a certain amount of MB or CuSO_4_.5H_2_O in DI water. In a typical adsorption test, dried beads (100 mg) were placed in 250 mL Erlenmeyer flasks containing 100 mL of the dye solutions (50 mg L^−1^, dilution of the stock solutions) at room temperature different adsorption pH values and times. The MB concentration was obtained by UV-VIS measurements (Hach, DR6000, Loveland, CO, USA) at 665 nm. The Cu (II) ion concentration was measured by atomic absorption spectrometry (AAS, Thermo Elemental, FS 95, λ 324.8 nm, Waltham, MA, USA). Equilibrium adsorption capacity per unit mass of the dried composite (q_e_, mg g^−1^) was calculated using Equation (1):(1)$$q_{e}=\frac{\left(C_{t}-C_{o}\right)V}{W}$$
where C_ο_ is the initial concentration of the MB or metal ions (mg L^−1^) and C_e_ is the concentration of pollutant remaining in the solution at equilibrium, V and W are the solution volume (L) and the composite mass (g), respectively. 

### 3.5. Adsorption Isotherms 

Langmuir and Freundlich’s models were applied to the experimental data to evaluate equilibrium characteristics in adsorption studies. The Langmuir model (Equation (2)) considers the molecular monolayer adsorption on the homogeneous adsorbent surface with similar sites. However, this model neglected the interactions between adsorbed molecules [37]. For the Freundlich model (Equation (3)), it assumes heterogeneous adsorption over independent sites with the non-uniform surface [38].
(2)$$\frac{C_{e}}{q_{e}}=\left(\frac{1}{K_{L}q_{m}}\right)+\left(\frac{1}{q_{m}}\right)C_{e}$$
(3)$${\mathrm{lnq}}_{e}={\mathrm{lnK}}_{F}+\left(\frac{1}{n}\right){\mathrm{lnC}}_{e}$$
where q_e_ and q_m_ are the adsorption capacity and maximum adsorption capacity at equilibrium (mg g^−1^), C_e_ is the concentration of MB or Cu^2+^ ions at equilibrium (mg L^−1^), K_L_ is the Langmuir adsorption constant (L mg^−1^), K_F_ is the Freundlich constant (mg g^−1^), and 1/n is the Freundlich constant related to the surface heterogeneity of the specific pollutant.

### 3.6. Adsorption Kinetics

The two models of adsorption kinetics, pseudo-first-order (Equation (4)) and pseudo-second-order (Equation (5)), were used to evaluate interactions between adsorbents and pollutants [4,33].
(4)$$\frac{{\mathrm{dq}}_{t}}{\mathrm{dt}}=k_{1}\left(q_{e}-q_{t}\right)\;\mathrm{or}\;\ln\left(q_{e}-q_{t}\right)={\mathrm{lnq}}_{e}-k_{1}t$$
(5)$$\frac{{\mathrm{dq}}_{t}}{\mathrm{dt}}=k_{2}{\left(q_{e}-q_{t}\right)}^{2}\;\mathrm{or}\;\frac{t}{q_{t}}=\frac{1}{k_{2}q_{e}^{2}}+\frac{t}{q_{e}}$$
where q_e_ and q_t_ are the adsorption capacity at the equilibrium (mg g^−1^) and at time t, k_1_ and k_2_ are the pseudo-first-order and pseudo-second-order rate constants (min^−1^), respectively, and t is the adsorption time (min).

### 3.7. Recycle Test

The initial MB and Cu(II) ions concentrations were 50 mg L^−1^ and 500 mg L^−1^, respectively. The adsorption time is 120 min. The pH values are 8 and 5 for MB and Cu(II) ions adsorption, respectively. After loading with MB or copper ions, the beads were separated by filtering and eluted with 0.1 M H_2_SO_4_ solution for regeneration and reused in the reusability study five times. The eluted solutions were analyzed for MB or Cu(II) ions.

## 4. Conclusions

The higher adsorption capacities of the beads for MB and Cu(II) ions were obtained when the higher porosities of the samples were used. The MB adsorption capacity is lower than the Cu(II) ions’ adsorption capacity due to the strong interaction of the positive metal ions with the negative-charge groups in the composite structure. The isotherm and kinetics models indicated that the adsorption of MB and copper on the bead is a suitable mechanism via multi-site interactions with a monolayer formation. Besides, the adsorbents can be quickly recovered as solid beads and particularly reused to remove metal ions and cationic dyes from wastewater. This work could also recommend using a chitosan-based nanocomposite for reducing other dyes and heavy metal ions in wastewater.

## Figures and Tables

**Figure 1 molecules-26-06127-f001:**
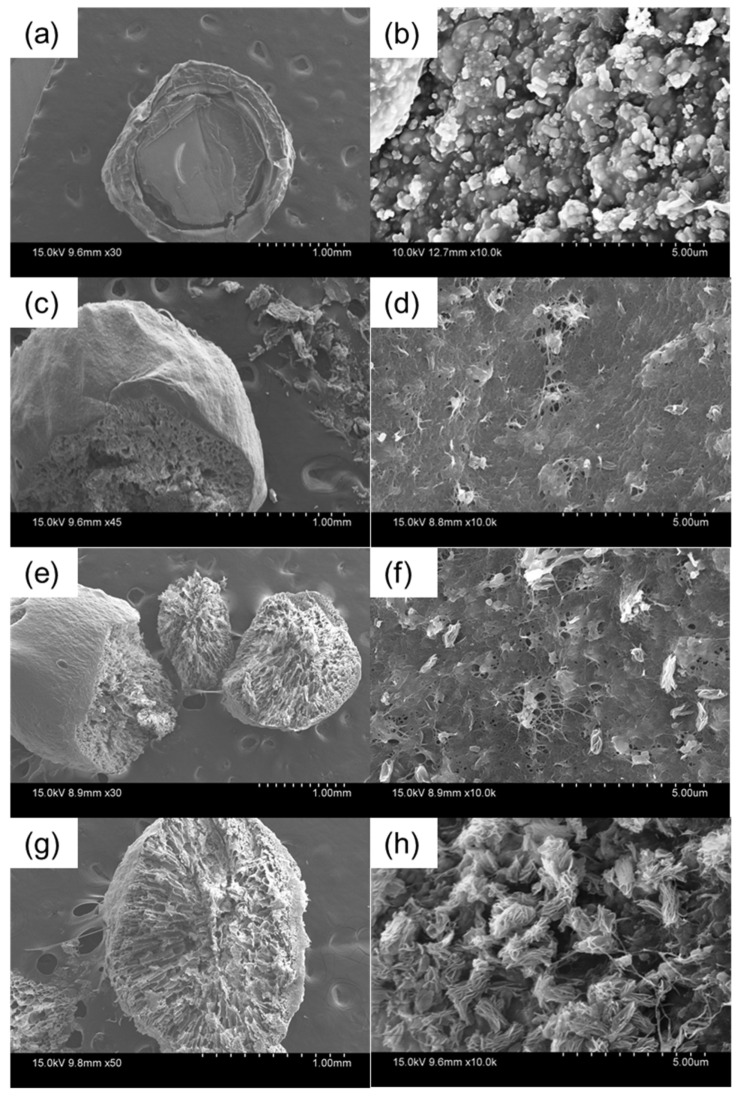
SEM images of (**a**,**b**) HGC-0, (**c**,**d**) HGC-1, (**e**,**f**) HGC-2, and (**g**,**h**) HGC-3 beads at different magnifications.

**Figure 2 molecules-26-06127-f002:**
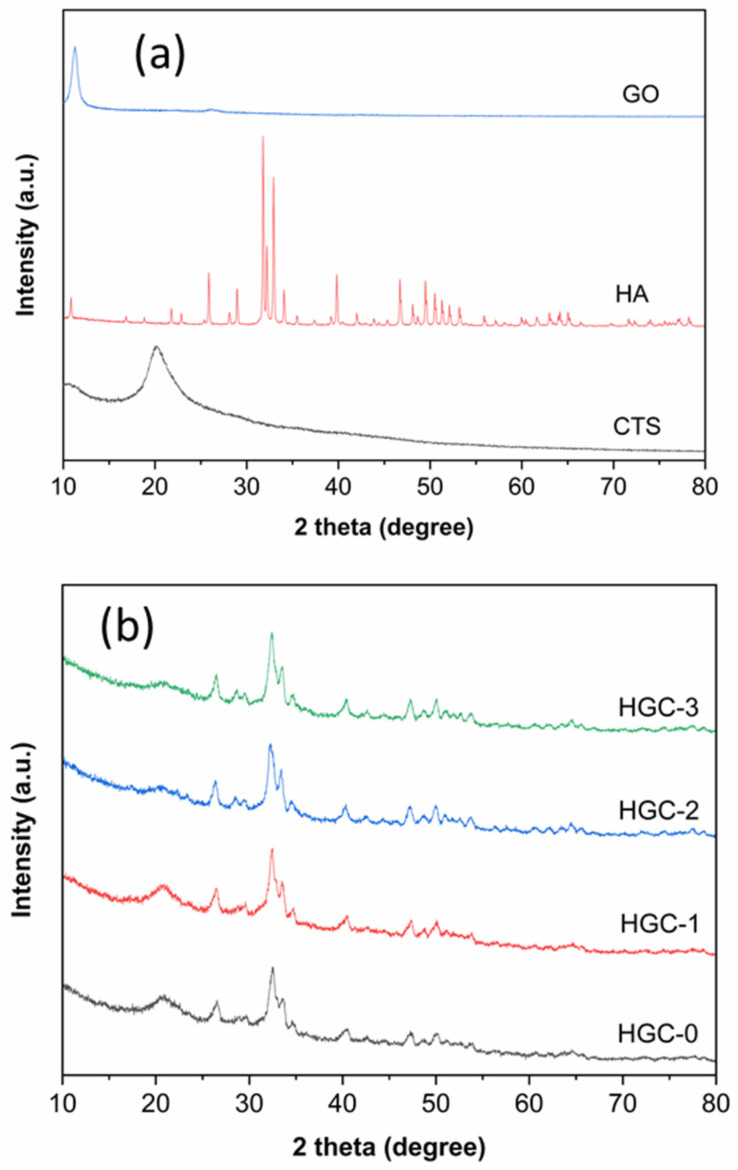
XRD patterns of (**a**) bare GO, pure HA, pure CTS, and (**b**) various HGC beads.

**Figure 3 molecules-26-06127-f003:**
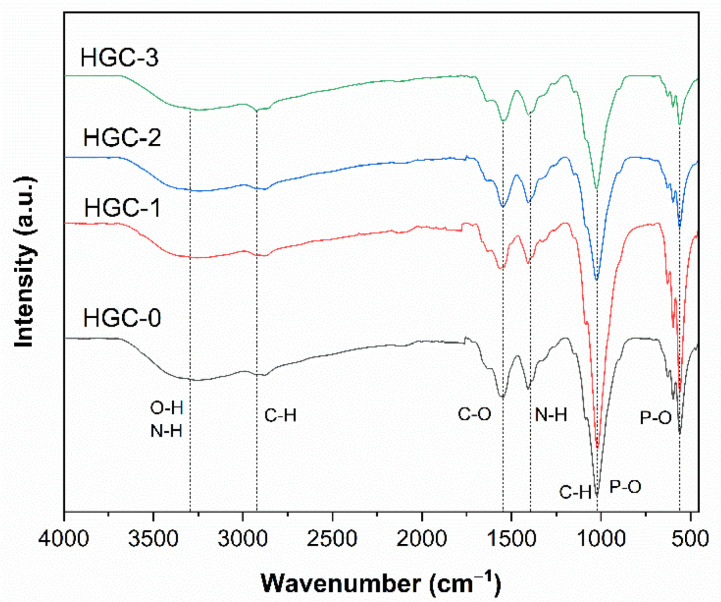
FTIR spectra of various HGC beads.

**Figure 4 molecules-26-06127-f004:**
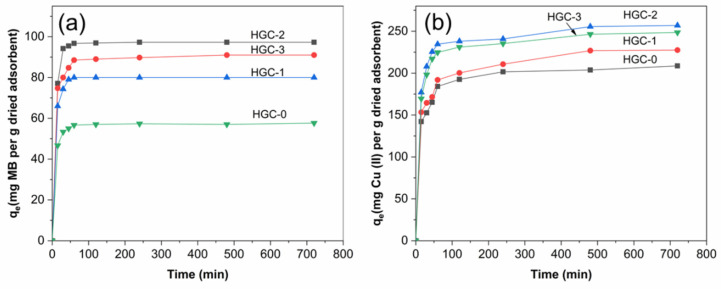
Effect of contact time on sorption capacities of the various HGC beads with (**a**) MB and (**b**) Cu (II) ions. Experimental conditions: 50 mg L^−1^ initial MB concentration at pH 8.0, 500 mg L^−1^ initial Cu (II) concentration at pH 5.0, and 100 mg dried beads under room temperature.

**Figure 5 molecules-26-06127-f005:**
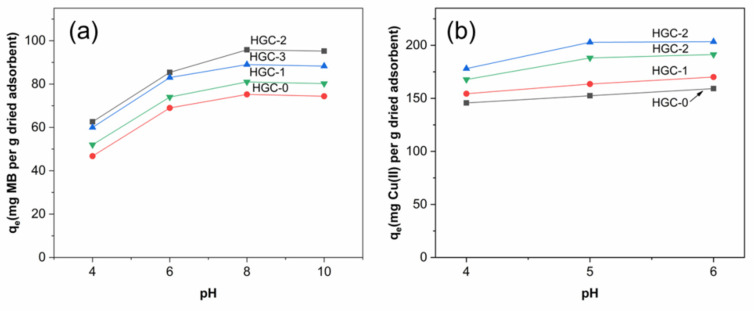
Effect of pH medium on sorption capacities of the various HGC beads with (**a**) MB and (**b**) Cu (II). Experimental conditions: 500 mg L^−1^ initial Cu (II) concentration, 50 mg L^−1^ initial MB concentration, and 100 mg dried beads under room temperature.

**Figure 6 molecules-26-06127-f006:**
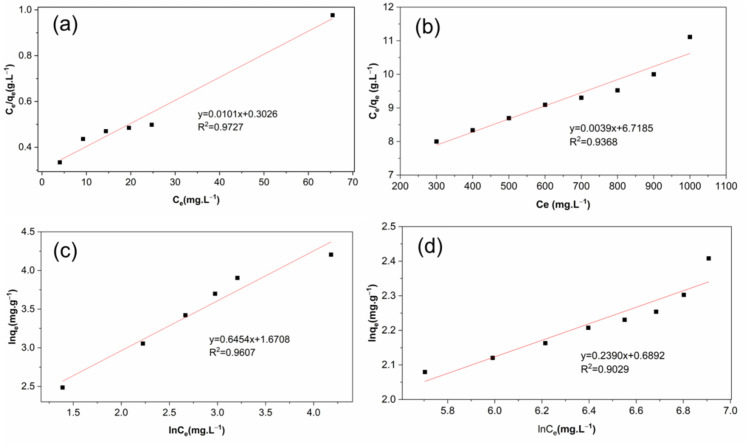
Fitting experimental data of Langmuir (**a**,**b**) and Freundlich (**c**,**d**) isotherms for (**a**,**c**) MB and (**b**,**d**) Cu (II) ions interaction to the HGC-2 beads. Experimental conditions: adsorption time of 120 min, 100 mg of dried HGC-2 beads, 100 mL solution, pH 8.0 (MB), or pH 5.0 (Cu(II) ions) under room temperature.

**Figure 7 molecules-26-06127-f007:**
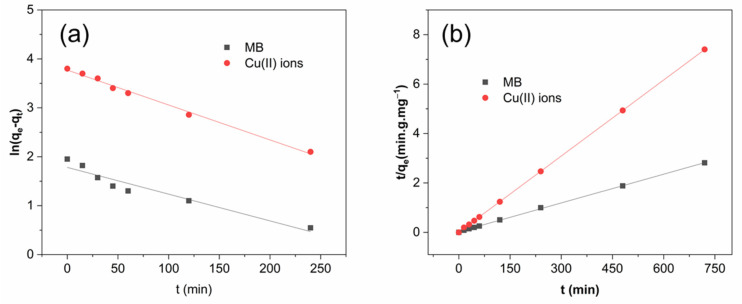
(**a**) Pseudo-second-order kinetic model and (**b**) pseudo-first-order kinetic model for MB and Cu (II) ions adsorption on the HGC-2 beads. Experimental conditions: 100 mg of dried HGC-2 beads, 100 mL solution, pH 8.0 (MB) or pH 5.0 (Cu(II) ions) under room temperature.

**Figure 8 molecules-26-06127-f008:**
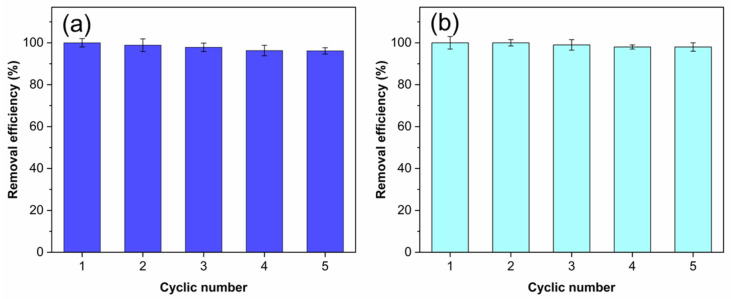
The removal efficiency of HGC-2 beads in sorption of (**a**) MB and (**b**) Cu (II) ions for five cycles. Experimental conditions: 50 mg of MB or 500 mg of Cu(II) ions, 100 mg of dried HGC-2 beads, 100 mL solution, pH 8.0 (MB), or pH 5.0 (Cu(II) ions) under room temperature.

**Table 1 molecules-26-06127-t001:** Surface and pore structural parameters of the different composites.

Sample	BET Specific Surface Area (m^2^ g^−1^)	BJH Specific Pore Volume (cm^3^ g^−1^)	BJH Pore Size (nm)
HGC-0	6.86	0.078	10.11
HGC-1	13.62	0.042	11.91
HGC-2	16.87	0.041	12.20
HGC-3	29.74	0.046	16.17

**Table 2 molecules-26-06127-t002:** Parameters of the fitting Langmuir and Freundlich isotherms for the adsorption of MB and Cu(II) ions onto the HGC-2 sample.

Isotherm Models	Isotherm Parameters	Adsorbate	
		MB	Cu(II) Ions
Langmuir	R^2^	0.9727	0.9368
	KL (L mg^−1^)	0.0333	0.0058
	qmax (mg g^−1^)	99.00	256.41
	χ^2^	0.0014	0.0721
	SD (%)	3.719	13.770
Freundlich	R^2^	0.9607	0.9029
	KF (L mg^−1^)	5.1364	1.9921
	n	1.5494	4.1841
	χ^2^	0.0190	0.0012
	SD (%)	3.519	26.845

**Table 3 molecules-26-06127-t003:** Kinetic parameters in the adsorption of MB and Cu(II) ions onto the HGC-2 sample.

Adsorbate	Pseudo-First-Order			Pseudo-Second-Order		
	k_1_ (min^−1^)	q_e_ (mg g^−1^)	SD (%)	k_2_ (g mg^−1^ min^−1^)	q_e_ (mg g^−1^)	SD (%)
MB	0.0149	54.93	6.9	9.2 × 10^−3^	94.46	1.26
Cu(II) Ions	0.0200	5.76	7.3	0.5 × 10^−3^	256.40	1.91

## Data Availability

Not applicable.

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
