# Peer review of "Highly Porous Hydroxyapatite/Graphene Oxide/Chitosan Beads as an Efficient Adsorbent for Dyes and Heavy Metal Ions Removal"

_molecules, 2021, doi:10.3390/molecules26206127_

Round 1
Reviewer 1 Report
The article is devoted to the removal of copper compounds and organic dye methylene blue from model water. Complex sorbents based on a mixture of hydroxyapatite/chitosan modified with graphene oxide additives were chosen as a material for the sorption of pollutants. The literature review (introduction) displays brief information about the preparation and use of similar materials, as well as traditional processes for water purification from heavy metal compounds. The research methods presented in the work are traditional for solving the assigned tasks and confirm the high degree of reliability of the results obtained. The results and their discussion clearly demonstrate not only the formation of sorbent layers but also provide data on the mechanisms of removal of various pollutants from aqueous media.
The main advantage of the presented work is the new idea of using a graphene oxide additive as an enhancer (activator) of a sorbent based on hydroxyapatite / chitosan. Considering the developed surface of graphene oxide, its use in the processes of sorption purification of various media can be very promising. It is necessary to note the high level of elaboration of the issue and analytical control. The use of modern high-precision equipment described in the materials allows you to be confident in the reliability of the data obtained.
As the main controversial point is the choice of copper ions as the object of study, which is perfectly removed from water by traditional methods of purification, such as neutralization, precipitation and ion exchange. The issue of desorption remains no less important, both from the point of view of the choice of the method / reagent / process conditions and from the point of view of the number of sorption-desorption cycles.
As a recommendation for continuing research, consider the proposal:
- The Research Methods (chapter 3) and Results (chapter 2) sections need to be reversed (currently in the text, results first).
- There is no information about the processes of desorption of pollutants from its surface (re-using the material)
- In conclusion, you write "The isotherm and kinetics models indicated that the adsorption of MB and cadmium on the bead". I'm sure this is a "mistake" and you wanted to write copper
- In further research, it would be great to see examples of sorption of other metals, such as nickel, or radionuclides and information about the effect of organic ligands (eg tartrate) on sorption efficiency. Copper compounds can give complex compounds even with ammonium and this sorption process may change dramatically
Author Response
The article is devoted to the removal of copper compounds and organic dye methylene blue from model water. Complex sorbents based on a mixture of hydroxyapatite/chitosan modified with graphene oxide additives were chosen as a material for the sorption of pollutants. The literature review (introduction) displays brief information about the preparation and use of similar materials, as well as traditional processes for water purification from heavy metal compounds. The research methods presented in the work are traditional for solving the assigned tasks and confirm the high degree of reliability of the results obtained. The results and their discussion clearly demonstrate not only the formation of sorbent layers but also provide data on the mechanisms of removal of various pollutants from aqueous media.
The main advantage of the presented work is the new idea of using a graphene oxide additive as an enhancer (activator) of a sorbent based on hydroxyapatite / chitosan. Considering the developed surface of graphene oxide, its use in the processes of sorption purification of various media can be very promising. It is necessary to note the high level of elaboration of the issue and analytical control. The use of modern high-precision equipment described in the materials allows you to be confident in the reliability of the data obtained.
As the main controversial point is the choice of copper ions as the object of study, which is perfectly removed from water by traditional methods of purification, such as neutralization, precipitation and ion exchange. The issue of desorption remains no less important, both from the point of view of the choice of the method/reagent/process conditions and from the point of view of the number of sorption-desorption cycles.
® Answer: Thank you very much for your good comments and suggestions. We have carefully checked and corrected the current work, and they must be considered in further studies. With your help, our works must be improved so much. Hopefully, the current version meets your request for publication in Molecules. In the revised manuscript, the corrections are highlighted in blue.
As a recommendation for continuing research, consider the proposal:
- The Research Methods (chapter 3) and Results (chapter 2) sections need to be reversed (currently in the text, results first).
® Answer: Thank you very much for your advice. We are sorry for keeping the order because the current sections are arranged based on the journal format.
- There is no information about the processes of desorption of pollutants from its surface (re-using the material)
® Answer: Thank you very much for your comment. The processes of desorption of pollutants from its surface were presented in Section 2.6 on pages 8 &9 and Section 3.7. on page 10.
- In conclusion, you write "The isotherm and kinetics models indicated that the adsorption of MB and cadmium on the bead". I'm sure this is a "mistake" and you wanted to write copper.
® Answer: We are sorry for our mistake. The word has been corrected. Thank you.
- In further research, it would be great to see examples of sorption of other metals, such as nickel, or radionuclides and information about the effect of organic ligands (eg tartrate) on sorption efficiency. Copper compounds can give complex compounds even with ammonium and this sorption process may change dramatically.
® Answer: You are right. Thank you very much. We must consider these excellent suggestions for our further research.

Reviewer 2 Report
The manuscript presents the synthesis of hydroxyapatite / graphene oxide / chitosan beads by a facile one-step method. The structure of the beads was examined by X-ray diffraction and infrared spectroscopy. The effect of the contact time, pH medium, dye / metal ion initial concentration, and recycle ability on the adsorption was investigated. The manuscript is well structured. I find that the manuscript contains new information and is suitable for publication after improve some important points in it.
I find that the supplementary file is appropriate, provides additional information needed by readers and does not require corrections.
- The discussion on infrared spectra of beads needs to be expanded. The authors have to explain the difference between the spectra of the samples with GO and without GO (HGC-0). They made the conclusion reached by the authors is "Overall, the beads' spectra supported that chitosan chains were successfully combined with GO sheets and HA particles." How is effect on the IR spectra?
- In the part which described the study of the effect of pH medium on sorption capacities, it should be explained why they consider in the case of MB that the maximum value is reached at pH = 8. Since at pH = 7 the medium changes its type from acidic to basic, how it affects . It is not clear the possibility to reach maximum sorption capacities at lower values pH < 8. As far as it is clear from the text, the explanation of this dependence is related to “The surface charge of HGC bead changed from positive to negative with the increase of pH from 4.0 to 10.0. “How was this result obtained or is it a citation from previous research?
- In cyclic tests, it must be clarified exactly how the absorbent is separated.
- Although there are clear abbreviations such as GO nanosheets, they should be specified in the text when they first appear.
Author Response
The manuscript presents the synthesis of hydroxyapatite / graphene oxide / chitosan beads by a facile one-step method. The structure of the beads was examined by X-ray diffraction and infrared spectroscopy. The effect of the contact time, pH medium, dye / metal ion initial concentration, and recycle ability on the adsorption was investigated. The manuscript is well structured. I find that the manuscript contains new information and is suitable for publication after improve some important points in it.
I find that the supplementary file is appropriate, provides additional information needed by readers and does not require corrections.
® Answer: Thank you very much for the very helpful comments and suggestions. We have carefully checked and corrected the current work as followings. With your help, our works must be improved so much. Hopefully, the current version meets your request for publication in Molecules. In the revised manuscript, the corrections are highlighted in blue.
- The discussion on infrared spectra of beads needs to be expanded. The authors have to explain the difference between the spectra of the samples with GO and without GO (HGC-0). They made the conclusion reached by the authors is "Overall, the beads' spectra supported that chitosan chains were successfully combined with GO sheets and HA particles." How is effect on the IR spectra?
® Answer: Thank you very much for your good comment. As per your suggestion, we have added the spectrum of GO in the supporting information and the discussion in more details on infrared spectra of beads with GO and without GO in the revised manuscript.
- In the part which described the study of the effect of pH medium on sorption capacities, it should be explained why they consider in the case of MB that the maximum value is reached at pH = 8. Since at pH = 7 the medium changes its type from acidic to basic, how it affects . It is not clear the possibility to reach maximum sorption capacities at lower values pH < 8. As far as it is clear from the text, the explanation of this dependence is related to “The surface charge of HGC bead changed from positive to negative with the increase of pH from 4.0 to 10.0. “How was this result obtained or is it a citation from previous research?
® Answer: Many thanks. It is very good comment. In this study, the HGC beads consist of three components (GO, CTS, HA). All of them contributed to the MB and Cu (II) adsorption because of their functional groups (-NH2, -COOH, -OH) and their high porous surface. For example, chitosan chains posse the lone pair on the amine and carboxylic groups. The amine groups are protonated partially or entirely at pH ≤ 6.5 (pKa of NH2 groups), forming -NH3+ cations, whereas the hydroxyl groups' protonation occurs at pH ≤ 4.7 (pKa of -COOH groups) [Int. J. Biol. Macromol. 2020, 155, 142–152; Molecules 2019, 24, 4205; Ind. Eng. Chem. Res. 2012, 51, 6862–6868]. The result was obtained from the experimental observation. As your suggestion, we have added more information and corrected some sentences in the revised manuscript on page 6. Thank you very much.
- In cyclic tests, it must be clarified exactly how the absorbent is separated.
® Answer: Thank you very much for your recommendation. We have added more details about the separation of the absorbent in the revised manuscript in Section 3.7 on page 10.
- Although there are clear abbreviations such as GO nanosheets, they should be specified in the text when they first appear.
® Answer: Many thanks for your suggestion. GO, HA, CTS have been specified in the text on page 1.
